# Enzymatic Hydrolysis of Salmon Frame Proteins Using a Sequential Batch Operational Strategy: An Improvement in Water-Holding Capacity

**DOI:** 10.3390/foods13091378

**Published:** 2024-04-29

**Authors:** Suleivys M. Nuñez, Pedro Valencia, Tamara Solís, Silvana Valdivia, Constanza Cárdenas, Fanny Guzman, Marlene Pinto, Sergio Almonacid

**Affiliations:** 1Escuela de Ingeniería Química, Pontificia Universidad Católica de Valparaíso, Valparaíso 2340025, Chile; 2Centro de Biotecnología Daniel Alkalay Lowitt, Universidad Técnica Federico Santa María, Valparaíso 2390136, Chile; pedro.valencia@usm.cl; 3Programa de Doctorado en Biotecnología, Pontificia Universidad Católica de Valparaíso/Universidad Técnica Federico Santa María, Valparaíso 2390123, Chile; tamara.solis@alumnos.usm (T.S.); silvana.valdivia@usm.cl (S.V.); 4Núcleo de Biotecnología Curauma (NBC), Pontificia Universidad Católica de Valparaíso, Valparaíso 2373223, Chile; constanza.cardenas@pucv.cl (C.C.); fanny.guzman@pucv.cl (F.G.); 5Departamento de Ingeniería Química y Ambiental, Universidad Técnica Federico Santa María, Valparaíso 2390123, Chile; marlene.pinto@usm.cl (M.P.); sergio.almonacid@usm.cl (S.A.)

**Keywords:** enzymatic protein hydrolysis, salmon frames, water-holding capacity, sequential batch, nitrogen extraction, byproduct recovery

## Abstract

The meat industry uses phosphates to improve the water-holding capacity (WHC) of meat products, although excess phosphates can be harmful to human health. In this sense, protein hydrolysates offer an alternative with scientific evidence of improved WHCs. Salmon frames, a byproduct rich in protein, must be processed for recovery. Enzymatic technology allows these proteins to be extracted from muscle, and the sequential batch strategy significantly increases protein nitrogen extraction. This study focused on evaluating the WHC of protein hydrolysates from salmon frames obtained through double- and triple-sequential batches compared to conventional hydrolysis. Hydrolysis was carried out for 3 h at 55 °C with 13 mAU of subtilisin per gram of salmon frames. The WHC of each hydrolysate was measured as the cooking loss using concentrations that varied from 0 to 5% (*w*/*w*) in the meat matrix. Compared with those obtained through conventional hydrolysis, the hydrolysates obtained through the strategy of double- and triple-sequence batches demonstrated a 55% and 51% reduction in cooking loss, respectively, when they were applied from 1% by weight in the meat matrix. It is essential to highlight that all hydrolysates had a significantly lower cooking loss (*p* ≤ 0.05) than that of the positive control (sodium tripolyphosphate [STPP]) at its maximum allowable limit when applied at a concentration of 5% in the meat matrix. These results suggest that the sequential batch strategy represents a promising alternative for further improving the WHC of hydrolysates compared to conventional hydrolysis. It may serve as a viable substitute for polyphosphates.

## 1. Introduction

In the food industry, phosphates are essential additives used in many processed foods to increase the stability of an emulsion and reduce the loss of water because they improve the texture, quality, and cooking performance [1]. The water-holding capacity (WHC) is decisive in these products since it is closely related to improving sensory aspects. In addition, reduced weight loss in meat products is linked to economic benefits. The demand for healthy food is a latent and significant problem for the food industry. The consumption of food products containing polyphosphates as additives has increased dramatically in recent years. An example of these additives is the incorporation of phosphates into many food products to improve the WHC, the intake of which is estimated to have increased 100% in the last twenty years, from 500 to 1000 mg/day [2,3]. Consequently, the phosphates in these products are reaching potentially concerning levels among consumers, surpassing the recommended daily allowance (RDA) of 700 mg/day for phosphate intake [4]. Clinical, in vitro, in vivo, and epidemiological studies have reported that ingredients such as phosphates can cause human disorders associated with hypertension, cardiovascular disease, and cancer when consumed excessively [5,6]. Therefore, it is essential to find natural replacement ingredients that can compensate for the lost product characteristics, which are unavoidable when removing these artificial ingredients. In this context, enhancing the WHC and other functional properties of muscle proteins can be achieved by adding protein hydrolysates, thus improving the abovementioned parameters [7].

In this sense, marine products like salmon benefit from techniques that enhance their properties and minimize weight loss. It is worth mentioning that Chile has a significant production of salmon, being the second-largest producer in the world, followed by Norway [8]. As a high-production industry, it yields a substantial amount of byproducts [9] that can be processed and reused, providing new marketable products. This paves the way for new markets and fosters a circular economy [10,11]. Salmon structure protein hydrolysates contain peptides of high functional quality, a substantial content of essential amino acids, and relevant bioactive properties [12,13,14]. Therefore, enzymatic protein hydrolysis has emerged as an efficient alternative to valorize this fish waste [15,16]. Utilizing hydrolysates from these byproducts as ingredients in food formulations would be advantageous for enhancing WHC [17].

It is essential to highlight that the WHC of protein hydrolysates has been studied for various protein sources from which they are derived. Several studies have analyzed the WHC of protein hydrolysates, including gelatin hydrolysate derived from bovine skin [17], proteins extracted from Roselle seeds [18], protein hydrolysates from seal meat [19], fish protein hydrolysates [20], and canola protein hydrolysates [21], among others.

According to a review by Kristinsson and Rasco [22], peptides must form hydrogen bonds between their hydrophilic side groups and water molecules to retain water molecules efficiently. Hydrolysis exposes hydrophobic groups to the surface and transforms hydrophobic groups into hydrophilic ones, generating COOH and NH_2_ termini. Thus, the peptides resulting from the hydrolysis of myofibrillar proteins are proportionally more polar, which increases their ability to form bonds with water and therefore the solubility of the protein compared to its native form. Although this applies to fish myofibrillar proteins, the hydrophobicity of globular proteins, such as those in milk, may increase upon hydrolysis due to exposure to nonpolar amino acid residues. Furthermore, the increasing presence of polar groups during enzymatic hydrolysis significantly influences the amount of water absorbed and the moisture sorption isotherm. Other studies [23] suggest that all water molecules are influenced, to a greater or lesser extent, by the forces exerted by the polar groups of proteins in a three-dimensional structure.

A recent study by Valencia and collaborators [24] proposed salmon frame hydrolysis using a two-phase sequential batch as an advantageous operational strategy. After separating the aqueous phase in the first batch, the remaining solid phase was subjected to a second hydrolysis stage. In this second phase, an increase in the production of α-amino groups and nitrogen extraction was observed, even without adding additional protease. Notably, the concentration of these groups and the amount of nitrogen extracted were directly related to the amount of protease initially added [24]. This method resulted in an 80% increase in protein nitrogen extraction compared to traditional hydrolysis. Previous research has shown that WHC increases with increasing amounts of soluble protein [25,26]. Additionally, increased DH has been related to improved WHC [17].

Despite these advances, a sequential batch strategy to obtain enzymatic hydrolysates with improved WHCs has not yet been implemented. Therefore, the present investigation aimed to compare the WHC of salmon frame protein hydrolysates obtained through this strategy with those obtained by conventional hydrolysis. This approach will allow the identification of the hydrolysate with the highest WHC and therefore improve the efficiency and profitability of the process. The findings will also help establish relationships between nitrogen extraction, DH, and WHC, facilitating a comprehensive evaluation of the process regarding efficiency, cost, and product quality. This research provides the opportunity to produce hydrolysates with high WHCs through a sequential batch strategy, potentially expanding their application in industries such as food, cosmetics, and pharmaceuticals and adding value to final products.

## 2. Materials and Methods

### 2.1. Materials

The main materials used in this study were salmon frames (SFs) donated by Fiordo-Austral (Puerto Montt, Chile), and the enzyme subtilisin from the commercial preparation Alcalase 2.4 L obtained from Novozymes (Bagsvaerd, Denmark). The salmon frames contained 16% (*w*/*w*) protein and 56% (*w*/*w*) moisture. Salmon fillets were purchased from a local market, Caleta Portales (Valparaíso, Chile). Analytical-grade reagents were used in all experiments.

### 2.2. Enzymatic Hydrolysis of SF Proteins

The hydrolysis process was carried out within a stirring vessel, where 1.5 kg of preground SF was hydrolyzed by subtilisin at a native pH of 6.5, without pH control, and at 55 °C in a thermoregulated bath. Subsequently, when the reactor reached 55 °C, 13 AU of subtilisin was added per kg SF, according to Valencia and collaborators [24]. The operation was carried out with a conventional hydrolysis configuration and double- and triple-sequential batch configuration, with an equal distribution of protease doses and reaction times between the sequential batches, as shown in Table 1.

Conventional hydrolysis consisted of one-step hydrolysis operated for 3 h under the abovementioned conditions. The double- and triple-sequential batch modalities involved two and three consecutive batches, respectively, operated for a total time of 3 h using a total enzyme dose of 13 mAU/g of SF for all sequential batches. Therefore, the same total operating time and protease dose were used for the sequential batches as for conventional hydrolysis.

Once the reaction time was over, the reaction mixture from sequential batch 1 (SB1) was centrifuged at 8500× *g* for 10 min at 4 °C in a Hettich Rotina 420 R centrifuge (Tuttlingen, Germany). Three phases were obtained, ordered from top to bottom: oil, aqueous, and insoluble phases (including bones). The insoluble phase of SB1 was loaded back into the reactor, and an equal proportion of water was added. Once the temperature stabilized, protease was added to initiate hydrolysis in the second sequential batch (SB2). The same procedure was carried out for the triple-sequential batch. In this case, the resulting insoluble phase of SB2 was loaded back into the reactor, and an equal proportion of water was added. Once the temperature stabilized, protease was added to initiate hydrolysis in the third sequential batch (SB3). The protease doses and operation times in SB1, SB2, and SB3 were distributed equally between the batches, as presented in Table 1. Samples were taken occasionally to analyze the reaction products. Finally, the soluble phases of each sequential batch were lyophilized individually, as well as the mixture of the soluble phases of each sequential batch, for subsequent analysis. A schematic of this procedure is shown in Figure 1.

### 2.3. Hydrolysis Characterization

The Kjeldahl method quantified the total nitrogen present in the SF and in the soluble phase of each batch. During hydrolysis reactions, time-lapse samples were immediately mixed with an equal volume of 10% trichloroacetic acid and centrifuged at 10,000× *g* for 10 min. Aliquots of the supernatant were taken to measure the concentration of the α-amino groups (α-NH) released during the reaction and the total nitrogen in the soluble phase. Free α-NH groups were quantified by the ophthaldialdehyde (OPA) method using serine as a standard [24]. The concentration of released α-NH was determined by the difference between the initial concentration of the α-NH group (at time zero) and each value. After the operation of each sequential batch, the reaction mixture was centrifuged at 8500× *g* for 10 min at 20 °C to separate the oil, soluble, and insoluble phases (including bones), and each phase was weighed.

Nitrogen extraction was determined as the nitrogen transferred from the SF to the soluble phase. It was calculated from the ratio between the total nitrogen in the soluble phase and the total nitrogen in the unhydrolyzed SF, as shown in Equation (1) [27].
(1)$$Nitrogen\;extraction\;\left(\%\right)=\frac{\;total\;nitrogen\;in\;soluble\;phase\;\left(g\right)}{total\;nitrogen\;in\;unhydrolyzed\;SF\;\left(g\right)}\times 100$$

The estimated peptide chain length (PCL) in the soluble phase was calculated by the total nitrogen/free α-NH ratio.

The degree of hydrolysis (DH′) was determined by the relationship between the number of released α-NH groups and the total nitrogen in the SF, as shown in Equation (2). Total peptide bonds were slightly overestimated from the total nitrogen content in SF, resulting in a slight underestimation of DH′. Each sample was analyzed in triplicate via the OPA method [27].
(2)$$DH^{\prime }\left(\%\right)=\frac{released\;free\;\alpha -NH\;groups\;in\;soluble\;phase\;\left(g\right)}{total\;nitrogen\;in\;the\;SF\;\left(g\right)}\times 100$$

### 2.4. Water-Holding Capacity (WHC)

The WHC of each hydrolysate was measured as the cooking loss. The procedures described by [17] were followed, although with certain modifications. Each freeze-dried hydrolysate was added to 15 g samples of ground salmon meat at different concentrations ranging from 0 to 5% (*w*/*w* meat). These mixtures were left to stand for 4 h and then subjected to heat treatment at 80 °C for 10 min. Subsequently, the samples were cooled with tap water. The WHC was determined by measuring the amount of water lost during cooking. The samples were subjected to centrifugation using special two-part tubes for quantification. These 50 mL tubes contained glass beads that formed a porous bed. As described previously, a plastic support with a paper filter was placed on this bed, where the muscle tissue sample was located [20]. The samples were centrifuged at 250× *g* for 10 min at 20 °C. Figure 2 shows a diagram of the tubes with the samples before and after centrifugation.

### 2.5. Statistical Analysis

The data are presented as the means ± standard deviations of two replicates for the hydrolysis experiments. The analyses (measurements) of the samples were carried out in triplicate. Statistical analyses were performed using analysis of variance (ANOVA). Multiple comparisons of means were made using the LSD Fisher test. A *p*-value ≤ 0.05 was considered significant. All calculations were performed with InfoStat/L 2020I software.

## 3. Results and Discussion

### 3.1. Characterization of the Enzymatic Hydrolysis of SF Proteins

Ground samples of 1.5 kg SF were hydrolyzed using 13 mUA/g SF subtilisin at 55 °C and an uncontrolled native pH of 6.5. Hydrolysis was carried out by a conventional method for 180 min, and double- and triple-sequential batch configurations were used. The protease doses and reaction times were distributed equally in each sequential batch, as detailed in Table 1. Due to the high water content of the SF (56%), a homogeneous mixture was achieved in the reactor without the need to add additional water in the first batch [24]. The hydrolysis curves of these experiments are presented in Figure 3a,b.

The curves corresponding to conventional hydrolysis and the first sequential batch (SB1) of the double- and triple-sequential batch modalities showed a typical release pattern of α-NH groups similar to that observed with other proteins and proteases [17,28,29]. However, the hydrolysis rate decreased markedly after 60 and 90 min for triple- and double-sequential batch hydrolysis, respectively. Compared with the conventional modality, SB1 showed a slightly lower release of α-NH groups after these periods. The sequential batches SB2 and SB3 presented considerably lower releases of α-NH groups. It is relevant to mention that although SB1 used 50% and 75% less enzyme in the double- and triple-sequential batch configurations, respectively, compared to conventional hydrolysis, the yields of the released α-NH groups were similar. These saturation patterns in terms of the protease dose and the behavior observed in the hydrolysis curve have already been documented in previous studies that investigated the hydrolysis of SF proteins using subtilisin [24].

This effect could be due to the availability of active sites on the substrate. However, in-depth knowledge is needed to understand the complexity of enzymatic protein hydrolysis. Therefore, a complete set of parameters was evaluated when the reaction was stopped: nitrogen extraction, DH’, and peptide size estimation. The extraction of nitrogen into the soluble phase is a parameter that reflects the productivity of the reaction. This value was determined according to Equation (1), and the results are shown in Figure 4.

The extraction of total nitrogen from the sequential batch (83 and 78% for the double- and triple-sequential batches, respectively) was significantly higher (*p* ≤ 0.05) than that obtained with conventional hydrolysis (37%), indicating that it is possible to extract more nitrogen from the insoluble phase to the soluble phase in sequential batches than in conventional hydrolysis. This is because the percentage of nitrogen extraction is directly proportional to the concentration of total amino groups and the soluble fraction. Similar results were observed in [24].

In addition to the characterization of the process, the DH′ and PCL were determined for each experimental condition. The DH′ was estimated from the ratio of free α-NH groups/total nitrogen (Equation (2)), considering that it should be calculated from the ratio of α-NH groups/total peptide bonds. The PCL corresponds to a characterization of the peptides by calculating the ratio of total nitrogen/free α-NH groups; both are quantified in the soluble phase. The DH′ and PCL results obtained under each reaction condition are shown in Figure 5a,b.

The results indicated that in the double- and triple-sequential batch hydrolysis, the DH’ decreased progressively from SB1 to SB2 and SB3, respectively. These values were lower than those obtained with conventional hydrolysis. However, the DH’ for SB1 + SB2 double-sequential batch hydrolysis and SB1 + SB2 + SB3 triple-sequential batch hydrolysis was comparable (*p* > 0.05) to that obtained with conventional hydrolysis. This means that the peptide compositions of SB1, SB2, and SB3 differed.

The inverse relationship between DH′ and PCL caused the opposite effect on PCL. The largest peptides were found in SB3 (Exp 7). These findings showed that the different operating configurations influence the characteristics of the protein hydrolysate and its possible effect on the functional properties. Both DH′ and PCL are molecular attributes that determine the functional properties of hydrolysates [24]. Thus, we have shown that hydrolysates with different characteristics in terms of these properties can be obtained by varying the operating conditions. Therefore, it is possible to adjust the reaction conditions to achieve the desired characteristics of the hydrolysate.

### 3.2. Effect of the Sequential Batch Operation on the WHC

The WHC of each hydrolysate obtained was measured as the cooking loss after heat treatment of the samples, followed by centrifugation. Although the maximum allowable amount of polyphosphates in foods is limited to 0.5% by weight [10], adding up to 5% by weight of STPP was considered in this study. As a result, the effects of the WHCs of both STPP and the hydrolysates of SF proteins can be observed in Figure 6. The results showed a decrease in cooking loss compared with the control, depending on the concentration of each ingredient analyzed. All ingredients had a significantly lower cooking loss (*p* ≤ 0.05) than that of STPP at its maximum limit allowed in the industry (0.5% by weight) when applied at a 5% concentration in the meat matrix.

As the ingredient concentration in the meat matrix increased, a marked variation in the WHC was observed for each hydrolysate. Furthermore, in the case of the hydrolysates (SB3 triple-sequential batch hydrolysis, SB2 triple-sequential batch hydrolysis, and SB1 + SB2 + SB3 triple-sequential batch hydrolysis), the cooking loss was comparable (*p* > 0.05) to that of the STPP in its maximum limit allowed in the industry when the hydrolysates were applied in the meat matrix at concentrations of 0.5% and 1%. The lowest cooking loss was achieved with SB1 + SB2 + SB3 triple-sequential batch hydrolysis at a concentration of 5% by weight in the meat, resulting in a 78% reduction in cooking loss compared to that of the control.

The hydrolysates obtained through the strategy of double-sequential batches (SB1 + SB2 double-sequential batch hydrolysis) and triple batches (SB1 + SB2 + SB3 triple-sequential batch hydrolysis) exhibited a significant reduction (*p* ≤ 0.05) in cooking loss when they were applied from 1% by weight in the meat matrix compared to the hydrolysate obtained by conventional hydrolysis. The improvement in WHC is attributed to the different compositions of peptides present in each hydrolysate and the balanced distribution of peptides with hydrophobic and hydrophilic residues. Furthermore, it is essential to consider the possible presence of synergistic effects between these factors, which have been documented in previous studies [30].

Therefore, these findings suggest that sequential batch hydrolysis, in contrast to conventional hydrolysis, emerges as a promising alternative to enhance the WHC of hydrolysates further. This allows the creation of new food ingredients of natural origin that are safe and beneficial for both industry and the health of consumers. It is important to note that the dose of enzymes and the total reaction time are the same for both operation configurations. As mentioned in other studies and demonstrated in the present work, this new operational configuration allows for modulating the functional properties of protein hydrolysates [24].

## 4. Conclusions

A study on the water-holding capacity of salmon frame protein hydrolysates was carried out using a sequential batch operational strategy. The sequential batch hydrolysis operational strategy showed variations in the degree of hydrolysis, the nitrogen extraction, and the characteristics of the peptides, which influenced the functional properties. Furthermore, hydrolysates obtained by sequential batches significantly reduced cooking loss compared to conventional hydrolysis, indicating improved water-holding capacity. These findings suggest that sequential batch hydrolysis allows the generation of a varied composition of peptides that promote water-holding capacity. This could be a promising alternative to improve water-holding capacity in protein hydrolysates, offering safer, sustainable, efficient, and natural ingredients for the food industry.

## Figures and Tables

**Figure 1 foods-13-01378-f001:**
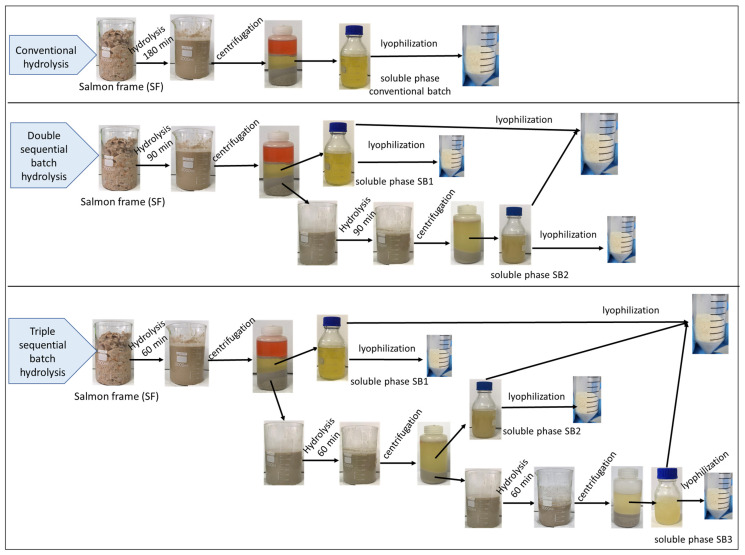
Schematic representation of the conventional hydrolysis process and that used in the double- and triple-sequential batch configuration of SF proteins for a total reaction time of 180 min at 55 °C, native and uncontrolled pH, and 13 AU of total subtilisin per kg SF.

**Figure 2 foods-13-01378-f002:**
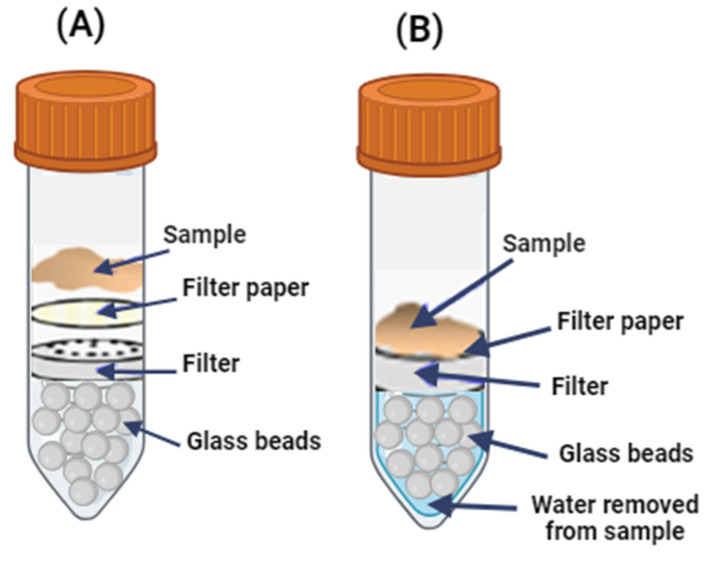
Diagram of the tubes used in the centrifugation of the samples for the determination of WHC. (**A**) Sample before centrifugation. (**B**) Sample after centrifugation.

**Figure 3 foods-13-01378-f003:**
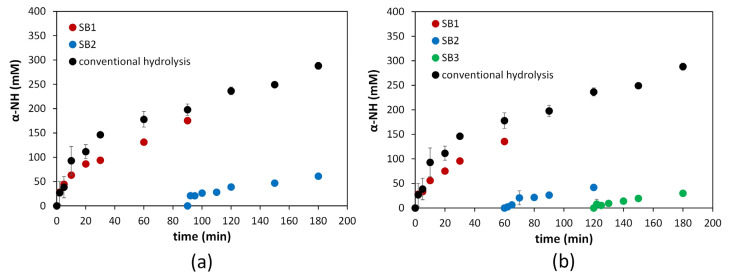
Reaction progress of the free α-NH group concentration (liquid samples) during the hydrolysis of SF in different operational configurations [conventional hydrolysis for 180 min, double- and triple-sequential batch hydrolysis for 180 min (90 and 60 min, respectively)]. (**a**) Comparison of the double-sequential batch method with conventional hydrolysis. (**b**) Comparison of the triple-sequential batch method with conventional hydrolysis. The reaction conditions were 55 °C, native pH (6.5) without a control, 100% SF, and 13 mAU/g SF. Each point is the mean of two experimental points, and the error bars are the standard deviations. SB1: sequential batch 1, SB2: sequential batch 2, SB3: sequential batch 3.

**Figure 4 foods-13-01378-f004:**
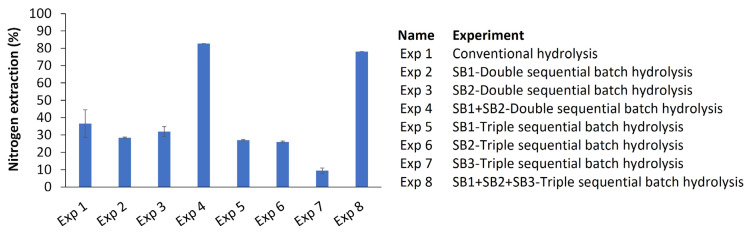
Nitrogen extraction after SF protein hydrolysis for different operating configurations [conventional hydrolysis for 180 min, double- and triple-sequential batch hydrolysis for 180 min (90 and 60 min, respectively)]. The experimental conditions were as follows: 100% SF, 13 mAU/g SF of subtilisin distributed in the indicated proportions between SB1/SB2/SB3, 180 min of total reaction time distributed as mentioned above, 55 °C, and an uncontrolled native pH of 6.5. Each point is the mean of two experimental points, and the error bars are the standard deviations. SB1: sequential batch 1, SB2: sequential batch 2, SB3: sequential batch 3.

**Figure 5 foods-13-01378-f005:**
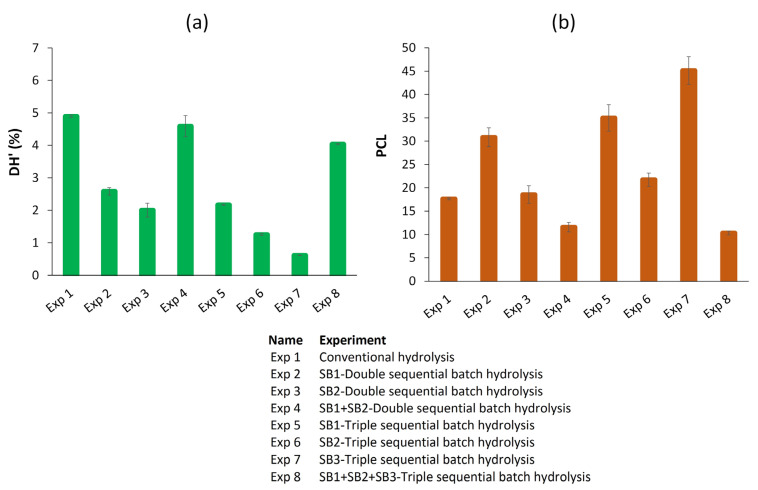
Characterization of the hydrolysates in the soluble phase at the end of the reaction at 55 °C for different operating configurations [conventional hydrolysis for 180 min, double- and triple-sequential batch hydrolysis for 180 min (90 and 60 min, respectively)]. (**a**) Degree of hydrolysis (DH′). (**b**) Peptide chain length (PCL). Each point is the mean of two experimental points, and the error bars are the standard deviations. SB1: sequential batch 1, SB2: sequential batch 2, SB3: sequential batch 3.

**Figure 6 foods-13-01378-f006:**
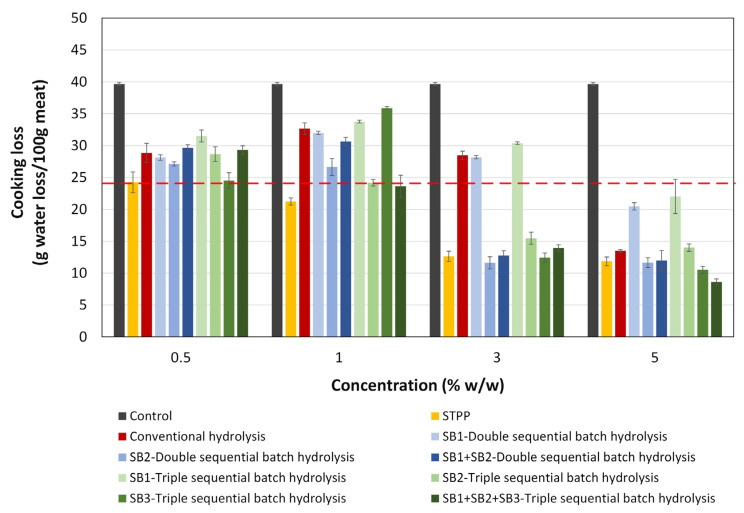
Cooking loss of salmon meat cooked with different amounts of SF hydrolysate in different operating configurations [conventional hydrolysis for 180 min, double- and triple-sequential batch hydrolysis for 180 min (90 and 60 min, respectively)]. Each value is expressed as the mean ± standard deviation of three replicates. The dashed line represents the STPP cooking loss for the application of the maximum concentration allowed in the industry, 0.5% (*w*/*w*).

**Table 1 foods-13-01378-t001:** Distribution of operation times and protease doses during the enzymatic hydrolysis of SF in conventional and sequential batch (SB) configurations.

Operation Configuration	Protease Dose (mAU/g SF)			Time (min)		
	SB1	SB2	SB3	SB1	SB2	SB3
Conventional hydrolysis	13			180		
Double-sequential batch hydrolysis	6.5	6.5	-	90	90	-
Triple-sequential batch hydrolysis	4.3	4.3	4.3	60	60	60

## Data Availability

The data presented in this study are available on request from the corresponding author due to privacy restrictions.

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
