# Peer review of "Enzymatic Hydrolysis of Salmon Frame Proteins Using a Sequential Batch Operational Strategy: An Improvement in Water-Holding Capacity"

_foods, 2024, doi:10.3390/foods13091378_

Round 1
Reviewer 1 Report
Comments and Suggestions for Authors
Comment 1:
The authors state in the description of the hydrolysis procedure (lines 117-144) that the sequential hydrolysis was performed by separating the solid residue after batch hydrolysis and mixing it with the same mass of water and adding the same amount of protease as in the first stage of the hydrolysis? Did the authors inactivate the protease after the first hydrolysis by incubating the reaction mixture in a boiling bath for some time or not? Is there a possibility that the remaining protease is from the first hydrolysis, so the units of enzyme activity/g substrate are higher than in the first hydrolysis? Did the authors try to add protease to the same reaction mixture after a certain time to check the course of hydrolysis?
Comment 2:
Why did you stop the conventional hydrolysis after 180 minutes if the – NH2 group content curve is increasing even after 180 minutes (Figure 3)?
Comment 3:
On what basis did you determine the time to add a new dose of protease in double and triple sequential hydrolysis? Did you follow a parameter or did you choose these time intervals at random?
Author Response
The authors would like to thank the reviewers for thoroughly assessing the manuscript. We have addressed all comments, and additions are marked in red font in the manuscript version that includes the changes. File attached.

Reviewer 2 Report
Comments and Suggestions for Authors
Overall, this manuscript is a very meaningful study. Protein hydrolysates from salmon frames are used to improve the water-holding capacity of meat products by replacing phosphates. However, there are some issues that have to be addressed before further consideration of the manuscript. And, my concern is whether the subtilisin used in the hydrolysis process will have a harmful effect on human health. In addition, the other specific comments are listed below:
Introduction
Please rewrite the first paragraph and the second paragraph into one paragraph. Phosphate should first introduce its function for the food industry, then explain the possible harm to human health, and finally explain the necessity of using protein hydrolysate to replace phosphate.
L54-55 Please delete “(and not most minorly)”.
L142-143 In the double and triple sequential batch, were the freeze-dried protein hydrolysates obtained in the SB1 and SB2 phases? The experimental results showed that the protein hydrolysates of these two phases were also determined. This description is somewhat confusing.
L155 “configuration of SF” Please revise the font size of this phrase.
L200 In general, the number of replicates required by statistics should be 3 or more.
L208 Did the authors study the optimum pH for hydrolysis by Alcalase 2.4 L from Novozymes? As far as I know, hydrolysis is better at pH under alkaline conditions. Why did the authors not adjust the pH of the reaction.
L215 Please clarify whether the determination of free α-NH group concentration is liquid protein hydrolysate or lyophilized protein hydrolysate.
L230 configurations. Please pay attention to the font size.
L264 According to the equations of PCL and DH, DH= 1/PCL. Why is there no inverse relationship between DH and PCL in Figure 5a and b?
L332 In figure 7a, firstly, please provide the P-value. Secondly, R2 was only 0.2019, it indicates a low fit.
L347 Please delete “water-holding capacity” and parentheses.
Comments on the Quality of English Language
No.
Author Response
The authors would like to thank the reviewers for thoroughly assessing the manuscript. We have addressed all comments, and additions are marked in red font in the manuscript version that includes the changes. File attached

Reviewer 3 Report
Comments and Suggestions for Authors
The manuscript presents the results of a general study of the benefits of enzymatic hydrolysis of salmon frame proteins using a sequential batch operational strategy. The authors compared the conventional enzymatic hydrolysis technology with its improved versions (double and triple sequential batch hydrolysis with different durations of individual stages). The results of the improvement were particularly considered in terms of increased water-holding capacity (WHC) of salmon frames protein hydrolysates. From a technological point of view, the presented solution has obvious advantages (including greater efficiency). For a complete assessment of the proposed solution, it would be necessary to compare the cost of the sequential batch hydrolysis operational strategy with conventional technology and its impact on the environment. Hence the request to the Authors to supplement the manuscript with these two issues (even in a general consideration).
Author Response

(The authors gave the same response as above.)

Round 2
Reviewer 2 Report
Comments and Suggestions for Authors
L39 “Therefore” should be replaced by “Because”.
L40, L46 the water-holding capacity (WHC) should be placed in the Line 40.
Comments on the Quality of English LanguageNo.
Author Response
The authors would like to thank the editor, and the reviewers for thoroughly assessing the manuscript. We have addressed all comments, and additions are marked in yellow in the manuscript version that includes the changes.
